# Applying a system dynamics approach for decision-making in software testing projects

**Wang Li, Chih-Chiang Fang**[ID]*

School of Computer Science and Software, Zhaoqing University, Zhaoqing, China

* peter@mail.sju.edu.tw

## Abstract

Enhancing software quality remains a main objective for software developers and engineers, with a specific emphasis on improving software stability to increase user satisfaction. Developers must balance rigorous software testing with tight schedules and budgets. This often forces them to choose between quality and cost. Traditional approaches rely on software reliability growth models but are often too complex and impractical for testing complex software environments. Addressing this issue, our study introduces a system dynamics approach to develop a more adaptable software reliability growth model. This model is specifically designed to handle the complexities of modern software testing scenarios. By utilizing a system dynamics model and a set of defined rules, we can effectively simulate and illustrate the impacts of testing and debugging processes on the growth of software reliability. This method simplifies the complex mathematical derivations that are commonly associated with traditional models, making it more accessible for real-world applications. The key innovation of our approach lies in its ability to create a dynamic and interactive model that captures the various elements influencing software reliability. This includes factors such as resource allocation, testing efficiency, error detection rates, and the feedback loops among these elements. By simulating different scenarios, software developers and project managers can gain deeper insights into the impact of their decisions on software quality and testing efficiency. This can provide valuable insights for decision-making and strategy formulation in software development and quality assurance.

## 1. Introduction

The stability and reliability of software play a crucial role in its success after development. Thorough testing of both ongoing and completed software is a crucial step in the software development process because it ensures high software quality. The responsibility of creating highly dependable software systems or more stable applications falls on the shoulders of the software development team. During the testing phase of the software development cycle, it is essential to allocate

**Data availability statement:** All relevant data are within the manuscript and its Supporting Information files.

**Funding:** This work was sponsored by the Guangdong Basic and Applied Research Foundation, China [grant number 2024A0505050040]. The funders had no role in study design, data collection and analysis, decision to publish, or preparation of the manuscript.

**Competing interests:** The authors have declared that no competing interests exist.

sufficient time and resources to uncover any hidden bugs. For customers purchasing the software, comprehensive, rigorous, and accurate testing of the software system is vital to enhance their confidence in using the product. From the perspective of the software development team, testing serves multiple purposes. It allows them to verify the correct execution of the programs they have written, assess whether the software's performance meets the required standards, and ensure that the system's security aligns with user requirements. Extending this, the testing phase also involves assessing the software's compatibility with various hardware and operating systems, evaluating its user interface for usability, and ensuring that it complies with legal and regulatory standards. Regular updates and maintenance testing are crucial to address any newly discovered vulnerabilities or bugs and to keep the software aligned with evolving user needs and technological advancements. Engaging with end-users for feedback during the testing phase can provide valuable insights into user experience and expectations, thereby guiding the development team in refining the software. Ultimately, the rigorous testing process not only enhances the reliability and stability of the software but also contributes to the reputation of the development team and the company. This promotes trust and satisfaction among users and stakeholders [1–3].

Historically, research in software testing and reliability has extensively focused on the development of software reliability growth models (SRGMs). These models are crucial for tracking the improvement of software reliability during extended testing periods. They serve a critical function in forecasting the evolution of software reliability over time, thereby aiding in the estimation of the total cost involved in software testing. This predictive capability empowers software developers to make informed decisions regarding the allocation of time and resources in software testing. It helps strike a balance between the costs of testing and debugging and the overall quality of the software. Over the past two decades, there has been substantial theoretical evolution in SRGMs. A prominent approach within these models is the Non-Homogeneous Poisson Process (NHPP), a widely adopted statistical method for modeling the software failure process. The core principle of NHPP is to treat the software failure rate function, denoted as $\lambda(t)$, as a function that is dependent on time. It assumes that $N(t)$, which represents the cumulative number of errors detected by time $t$, is a crucial metric. The function $N(t)$ is used to determine the number of program errors identified at any given time $t$ [4–11].

However, it is important to note that most conventional SRGMs operate under the assumption of "Perfect Debugging." This assumption simplifies the complexity of mathematical analysis by presuming that every identified bug is flawlessly fixed, which may not always reflect real-world scenarios. To further enhance this understanding, recent advancements in SRGMs are exploring more realistic scenarios. These considerations include the possibility of "Imperfect Debugging", where some errors may only be partially resolved or new issues may be introduced during the debugging process. Additionally, modern models are increasingly incorporating factors such as user feedback, varying testing environments, and the influence of software updates. There is also a growing emphasis on utilizing machine learning

techniques to enhance the predictive accuracy of these models. This involves considering complex datasets and real-world usage patterns [12–16].

Furthermore, some studies have been focusing on how SRGMs can be adapted to agile practices, where software development and testing are conducted in a more iterative and continuous manner. This involves developing models that can dynamically adjust to rapid development cycles and frequent code releases, which is a stark contrast to traditional models designed for longer, more static testing periods. Therefore, the issue of multiversion or multiphase software testing arises. Overall, the continuous evolution of SRGMs reflects an ongoing effort to align these models more closely with the evolving landscapes of software development and testing. The goal is to achieve a more accurate, dynamic, and practical approach to understanding and enhancing software reliability [17–19].

Recent advancements in software reliability and testing have highlighted the significance of robust models and strategies for optimizing software quality and performance. Pradhan et al. [20] emphasize the role of software reliability models and multi-attribute utility functions in determining optimal software release times, thereby providing a strategic framework for balancing reliability with development constraints. Building on this, Pradhan et al. [21] explore emerging trends in software reliability growth modeling, identifying future directions to enhance predictive accuracy and adaptability in complex software environments. In the context of debugging and fault management, Zada et al. [22] propose the use of Support Vector Machines (SVM) for classifying software failure incidents, demonstrating the potential of machine learning in improving debugging efficiency. Furthermore, Zada et al. [23] extend this approach by evaluating supervised machine learning classifiers for malware detection, highlighting the critical role of automated tools in identifying and mitigating software defects. Meanwhile, Ilyas et al. [24] address the challenges of software reliability in critical systems by presenting a fog computing-based architecture for pervasive health monitoring, which integrates robust testing and debugging mechanisms to ensure system resilience. Collectively, these studies underscore the evolving landscape of software testing, reliability, and debugging. They highlight the integration of advanced modeling techniques, machine learning, and innovative architectures to address the complexities of contemporary software systems.

"System Dynamics" is a methodological and analytical approach developed by Jay W. Forrester [25] at the Sloan School of Management at MIT. It emphasizes the construction of cause-and-effect relationships between variables through systematic thinking and feedback control theories. This approach is used to create simulation models, which are tools designed to simulate and project objectives based on interconnected relationships. Historically, system dynamics has primarily been employed to solve industrial and business management issues. This includes analyzing changes in production and employment, national economic trends, and corporate market strategies. Computer simulations play a crucial role in these applications by enabling the visualization of how various elements—such as system structure, policies, and time delays—interact and influence one another. In recent years, the scope of system dynamics has significantly expanded. It is now increasingly being applied to address environmental and social science challenges. Models based on system dynamics are constructed to understand and analyze the complex feedback relationships among factors such as industrialization, pollution, healthcare, and resource allocation. These models aim to identify the key factors that influence complex systems and suggest strategies for improvement [25–28].

In this study, in order to simplify the complexities often associated with mathematical derivations in traditional SRGMs, some of these models may rely on simplified assumptions or conditions. While this makes the models more manageable, it can lead to an incomplete portrayal of software reliability growth, potentially omitting crucial factors. This simplification may lead to inaccuracies in predicting software reliability and estimating testing costs. Addressing this issue, the current study adopts a system dynamics approach. This method involves mapping the impact of test and debug time on software reliability growth through a causal loop diagram, which enables system dynamics simulations. This approach differs significantly from traditional studies, which usually rely on differential equations to infer the mean value function of software reliability growth before calculating software reliability. The use of system dynamics in modeling allows for the consideration of more complex systems while avoiding the need for complicated mathematical derivations. The study's method

provides a more comprehensive and dynamic view of the software development process, considering various interconnected factors that traditional models might overlook.

To further elaborate on this, the study aims to integrate real-world scenarios into the system dynamics model, thereby providing a more realistic and practical perspective. This includes considering factors such as team capabilities, resource limitations, and market pressures, all of which can significantly impact the reliability of software and the testing processes. In summary, the study introduces an innovative framework for decision-making in software testing projects by applying a system dynamics approach, offering three key contributions to the field. First, the proposed methodology presents a comprehensive scientific management framework that integrates dynamic interactions among software testing, debugging processes, and resource allocation. Second, the model enhances practical relevance by incorporating a broader array of real-world factors, such as evolving team dynamics, imperfect debugging scenarios, and changing project constraints. Third, the study also accounts for the learning curve of testing and debugging personnel, allowing for a more accurate representation of improvements in software reliability throughout the testing process.

To summarize, these features bridge the gap between theoretical models and practical decision-making, empowering software testing managers to implement data-driven strategies that optimize software testing results.

## 2. Software reliability modeling and system dynamics

### 2.1 Basic model development

In the field of reliability engineering, the NHPP has emerged as a crucial tool for addressing reliability issues in various hardware and software scenarios. Unlike the Homogeneous Poisson Process (HPP), the NHPP is distinct in its ability to account for variable failure rates over time. This characteristic is particularly advantageous for SRGMs because the frequency of software defects typically decreases with continued testing and debugging. Prior to market release, software systems undergo extensive testing and debugging to ensure their quality. During this critical phase, software departments or companies evaluate multiple strategies, and decision-makers choose the most appropriate testing methods. The management's challenge is to find a balance between ensuring system stability and managing associated expenses. They also need to evaluate the impact of various resource allocations on the enhancement of system reliability. This includes assessing the effectiveness of testing and debugging efforts, as well as planning budgets for different resource scenarios.

Generally, the development of software reliability over time is modeled using mathematical functions in a counting process format, represented as $\{N(t), \ t \geq 0\}$. Within this framework, $N(t)$ is governed by a NHPP, which is defined by its mean value function, denoted as $M(t)$. The dynamics of this process can be mathematically described through a specific formulation.

$$P\left(N(t) = k\right) = \frac{[M(t)]^k e^{-M(t)}}{k!}, \ k = 0, 1, 2, \ldots; \ k \in N \tag{1}$$

Viewed from a different perspective, the mean value function essentially estimates the total number of errors detected and identified within a specific time period, starting from 0 and ending at a specific point in time $t$, and it can be presented as follows:

$$M(t) = \int_0^t \lambda(x) dx. \tag{2}$$

As the software undergoes testing, the number of remaining and undetected errors gradually decreases over time due to the debugging efforts. This reduction in undetected errors directly contributes to enhancing the reliability of the software. Based on this, the software reliability $R(t)$ can be defined as follows:

$$R(t) = e^{-(A-M(t))/\theta}, \tag{3}$$

where $A$ denotes the potential error number in a software system at the beginning and $(A - M(t))$ represents the number of remaining and undetected errors in a software system at time $t$. The parameter $\theta$ is an adjustment factor for software reliability. Additionally, as testing time approaches infinity, the software reliability metric is expected to progressively approach a value of one (expressed mathematically as $\lim_{t\to\infty} R(t) \to 1$).

Given the points discussed above, we assume that the occurrence of software errors in this study follows a NHPP, with any identified errors being resolved immediately. Consequently, the time required to fix errors is considered negligible. To develop a robust model for predicting software reliability, we incorporate the concept of the learning effect within the SRGM. Figure 1 presents a causal loop diagram that visualizes the software testing process within the context of the system dynamics model. In this causal loop diagram, the variable $\alpha$ represents the inherent ability of testing staff to independently detect software errors. This ability is based on their skills and training, rather than on insights gained from previous error patterns. This factor is distinct from the learning factor, $\beta$, which reflects the staff's ability to learn from past errors and improve with increased testing time. Therefore, the learning factor is related to the accumulated number of detected errors. Furthermore, the effect of the two factors is dependent on the scale of the testing staffs ($K$). Besides, the number of undetected errors at the beginning ($A - M(0)$) is the number of initial errors ($A$). However, some new errors may be introduced due to the staff's negligence and inappropriate debugging process. Therefore, in order to obtain a more accurate estimation, it is assumed that the software system may experience an increase in the proportion of new errors during the debugging process. It should be noted that there are the two flows in Figure 1. The right one represents the error detection per unit time ($\lambda(t) = \frac{dM(t)}{dt}$), and the left one represents the increase of the new errors due to imperfect debugging.

Drawing from the principles of the causal loop diagram, the error detection per unit time $\lambda(t)$ can be formulated through the use of differential equation techniques as follows:

$$\lambda(t) = \frac{dM(t)}{dt} = (A - M(t))\left(\alpha + \beta\left(\frac{M(t)}{A}\right)\right). \tag{4}$$

In order to apply the integration of natural logarithm, the above equation can be rearranged as follows:

$$\frac{\frac{dM(t)}{dt}}{A - M(t)} - \frac{\frac{dM(t)}{dt}}{\alpha + \left(\frac{\beta}{A}\right)M(t)} = 1. \tag{5}$$

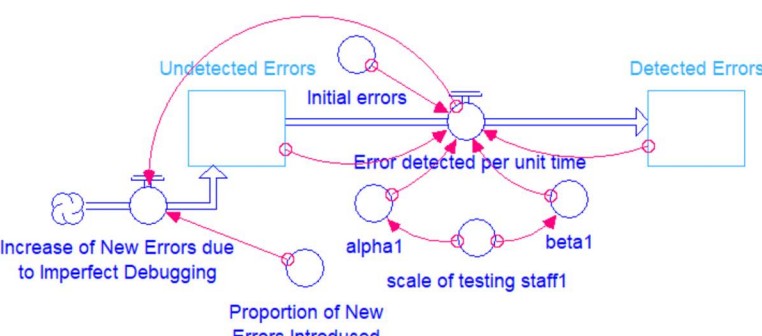

**Fig 1. Causal Loop Diagram of a Basic System Dynamics for the Software Testing Process.**

After the integration of the both sides of Equation (5), the result can be obtained as follows:

$$\int \left( \frac{\frac{dM(t)}{dt}}{(A-M(t))} - \frac{\frac{dM(t)}{dt}}{\alpha + \left(\frac{\beta}{A}\right)M(t)} \right) dt = \int 1\,dt$$

$$\rightarrow \frac{-\ln(-A+M(t)) + \ln(A\alpha + \beta M(t))}{\alpha+\beta} = t + constant. \tag{6}$$

In order to derive the mathematical form of $M(t)$, one must initially solve Equation (6) with respect to $M(t)$. This process results in a representation of $M(t)$ that includes an undetermined constant, as shown in the following expression:

$$M(t) = \frac{A(e^{(\alpha+\beta)(t+constant)} + \alpha)}{e^{(\alpha+\beta)(t+constant)} - \beta}. \tag{7}$$

Considering the initial condition of $M(0) = 0$, which implies that no errors are identified at the beginning of the testing period ($t = 0$), Equation (7) can be resolved by utilizing this initial condition to eliminate the unknown constant. As a result, Equation (7) is restructured as follows:

$$M(0) = \frac{A\left(e^{(\alpha+\beta)constant} + \alpha\right)}{e^{(\alpha+\beta)constant} - \beta} = 0. \tag{8}$$

After solving Equation (8) to find the unknown constant, the solution is obtained as follows:

$$constant = \frac{-\ln(\alpha)}{\alpha + \beta}. \tag{9}$$

Substituting this constant into Equation (7) allows us to derive the complete expression for $M(t)$ as:

$$M(t) = \frac{A\left(e^{(\alpha+\beta)\left(t - \frac{\ln(\alpha)}{\alpha+\beta}\right)} + \alpha\right)}{e^{(\alpha+\beta)\left(t - \frac{\ln(\alpha)}{\alpha+\beta}\right)} - \beta}. \tag{10}$$

This equation represents the initial form of the mean value function, which is utilized to estimate the average cumulative number of detected errors. Given that $\ln(\alpha)$ equals $\alpha$, the expression for $M(t)$ can be further simplified to:

$$M(t) = \frac{A\alpha(e^{(\alpha+\beta)t} - 1)}{\alpha e^{(\alpha+\beta)t} + \beta}. \tag{11}$$

For software managers aiming to assess the number of errors detected at a specific time t during the testing phase, it is crucial to understand the intensity function of the mean value, $\lambda(t)$. This can be found by computing the first derivative of $M(t)$, which obtains the mathematical form of $\lambda(t)$ as:

$$\lambda(t) = \frac{dM(t)}{dt} = \frac{A\alpha(\alpha + \beta)^2 e^{(\alpha+\beta)t}}{\left(\beta + \alpha e^{(\alpha+\beta)t}\right)^2}. \tag{12}$$

To effectively monitor the evolving rate of error detection per error at time t, the form of the error detection rate can be derived as follows:

$$D(t) = \frac{\lambda(t)}{A - M(t)} = \frac{\alpha(\alpha + \beta)e^{(\alpha+\beta)t}}{\beta + \alpha e^{(\alpha+\beta)t}}.$$

(13)

Notably, the error detection rate is a strictly increasing function, indicating that the efficiency of debugging is expected to improve as testing time advances. Moreover, when considering the scenario of imperfect software debugging, the number of initial errors may increase over testing time. Consequently, the initial errors can be expressed as a function of testing time, denoted as $A(t)$. The mathematical form of $A(t)$ can be defined based on the specific characteristics and dynamics of the software testing process.

Furthermore, the mathematical formulation of the mean value function is utilized to estimate crucial parameters' value for a specific testing team, assisting in the evaluation of their software testing effectiveness. Multifaceted testing environments present challenges, including collaboration among multiple testing groups, fluctuating learning factors, imperfect debugging, and various stochastic elements. These complexities create scenarios where traditional mathematical models struggle to apply. As a result, achieving accurate estimations becomes difficult due to the dynamic and interdependent nature of these factors. The fundamental mathematical model retains its usefulness in simpler testing situations. However, for more complex scenarios, a system dynamics approach is more appropriate. This method offers a comprehensive framework for understanding and analyzing the interactions and feedback loops among various elements in complex testing environments. It allows for the integration of various factors, such as team dynamics, different skill levels, changing software requirements, and resource limitations. By adopting this approach, it becomes possible to simulate and predict the behavior of the testing process under various conditions.

The next section will introduce how to estimate the parameter values of the fundamental mathematical model.

## 2.2. Parameter estimation

This study utilizes two main methods for parameter estimation in the proposed model: Maximum Likelihood Estimation (MLE) and Least Squares Estimation (LSE). These techniques are crucial in evaluating the effectiveness and precision of the model. Utilizing MLE and LSE allows for a comprehensive analysis of the collected software failure data, providing a solid foundation for benchmarking the proposed model against existing ones. The focus here is on the quality of fit, which is critical in determining how closely the new model reflects the actual patterns of software failures. This is a key aspect in validating the model's predictive accuracy.

(1) The MLE method is commonly used to estimate parameters in a presumed probability distribution, which is particularly relevant in an NHPP context. In this scenario, the likelihood function is tailored to reflect the unique characteristic of a time-varying event rate in the NHPP. MLE seeks to find the most probable parameter values within the distribution that best explain the observed data using the NHPP. The likelihood function is expressed as follows:

$$\mathcal{L}(A, \alpha, \beta) = Pr\{N(t_1) = M_1, N(t_2) = M_2, ..., N(t_n) = M_n\}$$

$$= \prod_{i=1}^{n} \frac{(M(t_i) - M(t_{i-1}))^{(M_i - M_{i-1})} (e^{-(M(t_i) - M(t_{i-1}))})}{(M_i - M_{i-1})!}.$$

(14)

By applying the natural logarithm to this equation and then calculating the first-order derivatives with respect to each parameter, and setting them to zero, we can solve the log-likelihood function using numerical methods. Particularly, if

an error-seeding method is employed to obtain an initial estimate of potential errors ($A$), estimating other parameters becomes more feasible.

(2) The LSE method minimizes the sum of squared differences between observed and predicted data in order to estimate model parameters. To implement LSE, a dataset comprising n pairs of observed values is utilized, represented as $(t_0, M_0), (t_1, M_1), (t_2, M_2), (t_n, M_n)$. In this context, each $M_i$ signifies the accumulated number of errors detected within the time frame $[0, t_i]$. The calculations for this estimation involve a specific procedure, tailored to align the model closely with the observed data points.

$$\text{Min } \boldsymbol{Er}(A, \alpha, \eta_0) = \sum_{i=1}^{n} (M_i - M(t_i))^2.$$

(15)

The estimation process involves solving the first-order derivatives of the error function with respect to each parameter and setting them equal to zero. Numerical methods are used for this purpose, and error seeding can simplify the estimation of $\alpha$ and $\beta$ by providing an initial estimate of potential errors ($A$).

Both methods are essential for understanding the rate and pattern of error detection in software testing, enabling managers to effectively navigate the testing phase and enhance software reliability.

## 2.3. Model verification and comparison

In this section, we assess the effectiveness of various SRGMs based on their fitness for different datasets obtained from related references. The evaluation begins by applying the LSE method to estimate the parameters of the proposed model

**Table 1. References of datasets.**

| Dataset | References | Dataset |
|---|---|---|
| Dataset 1 | Zhang and Pham [4] | Failure data of Misra system |
| Dataset 2 | Shyur [14] | Failure data of Misra system |
| Dataset 3 | Pham and Zhang [5] | Failure data of Tandem software |
| Dataset 4 | Zhang and Pham [7] | Failure data of Telecommunication system |

**Table 2. Summary of mean value function mean value function for the SRGMs.**

| SRGMs | M(t) | Characteristic |
|---|---|---|
| **Yamada [12]** | $M(t) = \left(\dfrac{Ab}{\alpha + b}\right)(e^{\alpha t} - e^{-bt})$, <br> $A(t) = Ae^{\alpha t}$ | This model is capable of exhibiting S-shaped curves characteristic and imperfect debugging issues. |
| **Huang [6]** | $M(t) = A\left(1 - e^{-rW^*(t)}\right)$, <br> $W(t) = \dfrac{N}{\sqrt[\kappa]{1 + Ae^{-\alpha\kappa t}}}$, <br> $W^*(t) = W(t) - W(0)$ | It also considers the testing-effort function and change-point issue. |
| **Pham and Zhang [5]** | $M(t) = \dfrac{1}{1 + \beta e^{-bt}}\left((A + c)\left(1 - e^{-bt}\right) - \dfrac{Ab}{b - \alpha}\left(e^{-\alpha t} - e^{-bt}\right)\right)$, <br> $A(t) = c + a(1 + e^{-\alpha t})$ | It incorporates a test-coverage function, as well as addressing the issue of imperfect debugging. |
| **Proposed model** | $M(t) = \dfrac{A\alpha^{((\alpha+\beta)t} - 1)}{\alpha^{(\alpha+\beta)t} + \beta}$ | It considers the inherent ability factor and the learning factor of testing staffs during the debugging process. |

using four specific datasets which are presented in Table 1. A comparative analysis is then conducted between the proposed model and three other classic SRGMs, as shown in Table 2.

For a comprehensive evaluation, two widely recognized indicators are employed:

(1) Mean Square Error (MSE): This measures the discrepancy between the estimat207.999 pted and actual values. The MSE formula is $MSE = \frac{\sum_{i=1}^{n}(M_i - M(T_i))}{n-k}$, where $M_i$ is the actual cumulative number of detected errors up to $M_i$; $M(T_i)$ is the estimated cumulative number of errors up to $T_i$; $n$ is the number of observations; and $k$ is the number of parameters in the model.

(2) R-squared (Rsq): This indicates how well the model accounts for the variability in the data. A higher Rsq value suggests a better fit. It's calculated as $Rsq = 1 - \frac{\sum_{i=1}^{n}(M_i - M(T_i))}{\sum_{i=1}^{n}(M_i - \frac{1}{n}\sum_{i=1}^{n} M_i)}$.

Table 3 demonstrates the superior performance of the proposed model in terms of MSE and R-squared. It should be noted that incorporating more parameters can enhance a model's flexibility and fitting capability, contingent on careful model design. The proposed model includes three parameters, allowing it to effectively adapt to different testing scenarios and accurately fit the test data. Figure 2 illustrates the excellent fit of the proposed model with the traditional CI (Confidence Intervals) to these artificially generated datasets, highlighting its adaptability and accuracy in various testing scenarios.

## 3. System dynamics model for software testing process

In this section, we utilize a system dynamics model to demonstrate a software testing process that involves multiple teams of testing staff. Figure 3 presents a conceptual framework that illustrates the interrelationships and feedback loops between various components involved in software testing and debugging.

In traditional models, all testing teams are treated as a single entity, and the parameters of the software reliability growth model are estimated based on this collective unit. However, in practice, testing teams often undergo changes and reorganization. In such cases, the model parameters previously estimated for the unified team are no longer applicable and must be re-estimated, as the values of these parameters have shifted. Furthermore, when considering multiple testing teams and the possibility of imperfect testing introducing new software defects, it becomes challenging to estimate testing efficiency using past mathematical inference methods. This is because deriving a closed-form mathematical model under these conditions is highly complex. Please refer to Figure 3, which illustrates that each testing team's testing efficiency can be described using a mathematical model. However, when considering the scenario of imperfect debugging across all teams, the combined mathematical models do not yield a closed-form expression to fully characterize the entire situation. To address this limitation, we utilize a system dynamics model, which allows for a comprehensive and dynamic representation of the entire testing process.

**Table 3. Comparisons of different fitting criteria.**

| Comparison Indicator | Datasets | Yamada [12] | Huang [6] | Pham and Zhang [5] | Proposed model |
|---|---|---|---|---|---|
| **MSE** | Dataset 1 | 27.42 | 32.05 | 38.67 | 34.83 |
| | Dataset 2 | 13.76 | 16.32 | 15.38 | 14.97 |
| | Dataset 3 | 8.56 | 13.38 | 10.53 | 9.13 |
| | Dataset 4 | 49.53 | 15.02 | 9.95 | 9.01 |
| **R-squared** | Dataset 1 | 0.974 | 0.973 | 0.966 | 0.971 |
| | Dataset 2 | 0.991 | 0.989 | 0.990 | 0.990 |
| | Dataset 3 | 0.991 | 0.988 | 0.990 | 0.991 |
| | Dataset 4 | 0.962 | 0.990 | 0.993 | 0.994 |

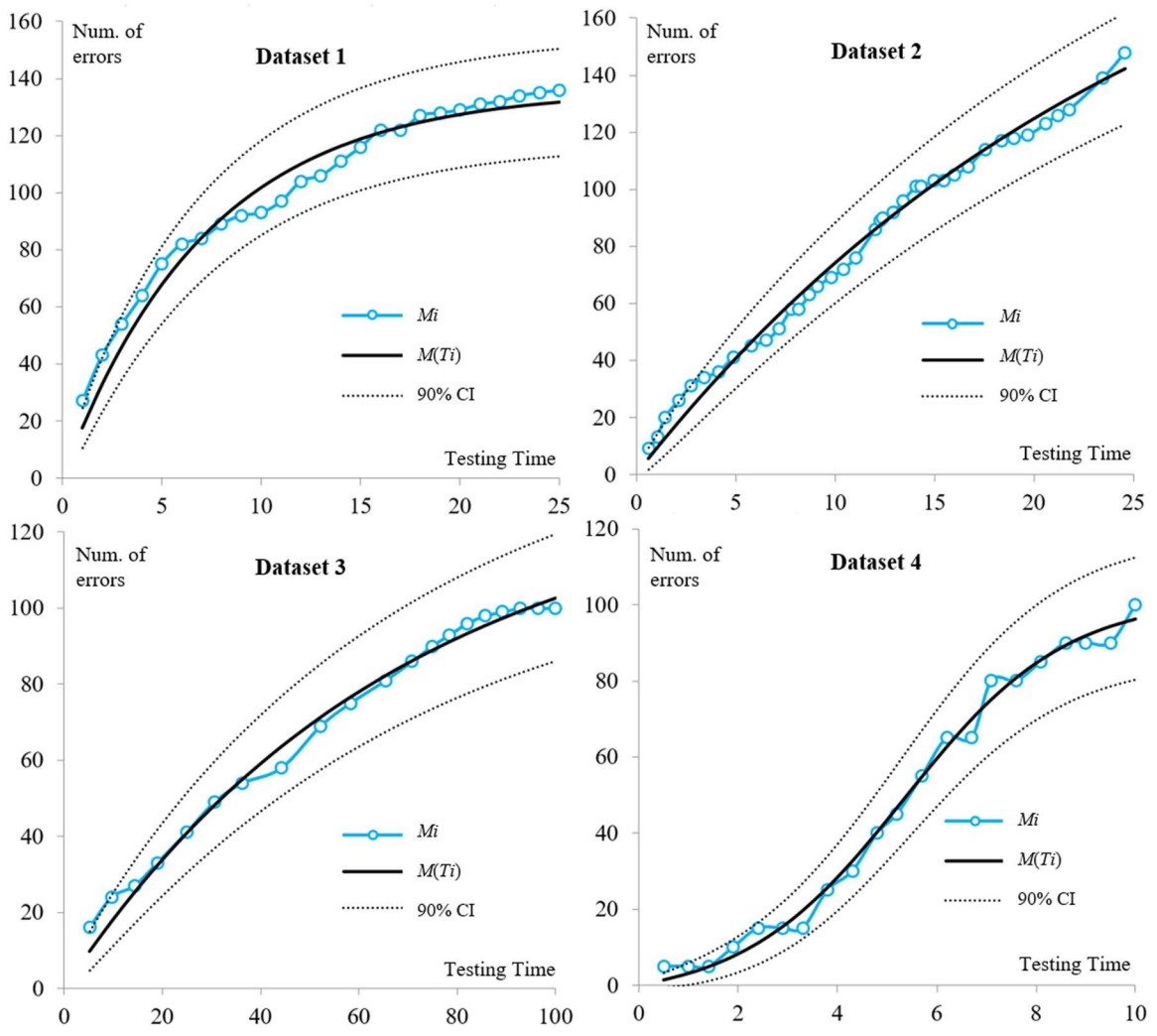

**Fig 2. Results of Fitness in Different Datasets for the Proposed Model (90% Confidence Interval).**

Here is a detailed description of the elements typically found in such a model:

(1) Testing Staff Teams: There are three teams ($p$: 1, 2, 3) involved in the testing process, each with different scales of staffing ($\xi_p$) due to varying human resource allocations. Since individual testing staff team have different education and work experience, their inherent testing ability factor ($\alpha_p$) and learning factor ($\beta_p$) will be also different. Therefore, the mean value function for the testing staff team $p$ will be $M_p(t) = \frac{A(t)\xi_p\alpha_p(\xi_p(\alpha_p+\beta_p)t-1)}{\xi_p\alpha_p\xi_p(\alpha_p+\beta_p)t+\xi_p\beta_p}$. Furthermore, the sum of all the mean value functions for the different teams is considered the stock "Detected Errors" in this system dynamics model. However, the learning factor may change over time due to testing, and can be represented as $\beta_p(t) = \beta_{p0} + \beta_{p1}t$. The parameters $\beta_{p0}$ and $\beta_{p0}$ represent the intercept and linear coefficient of the learning factor function.

(2) Error Detection per Time Unit: It is a flow in the causal loop diagram. This node aggregates the error detection rates from the three testing staff teams and demonstrates how it impacts the number of detected errors. It can be represented as $\sum_p \lambda_p(t)$. Since the velocity of the error detection will fluctuate in practice, the randomness of the flow is set as $\lambda_p(t) \sim N(\mu_{\lambda_p(t)}, \sigma_{\lambda_p(t)})$.

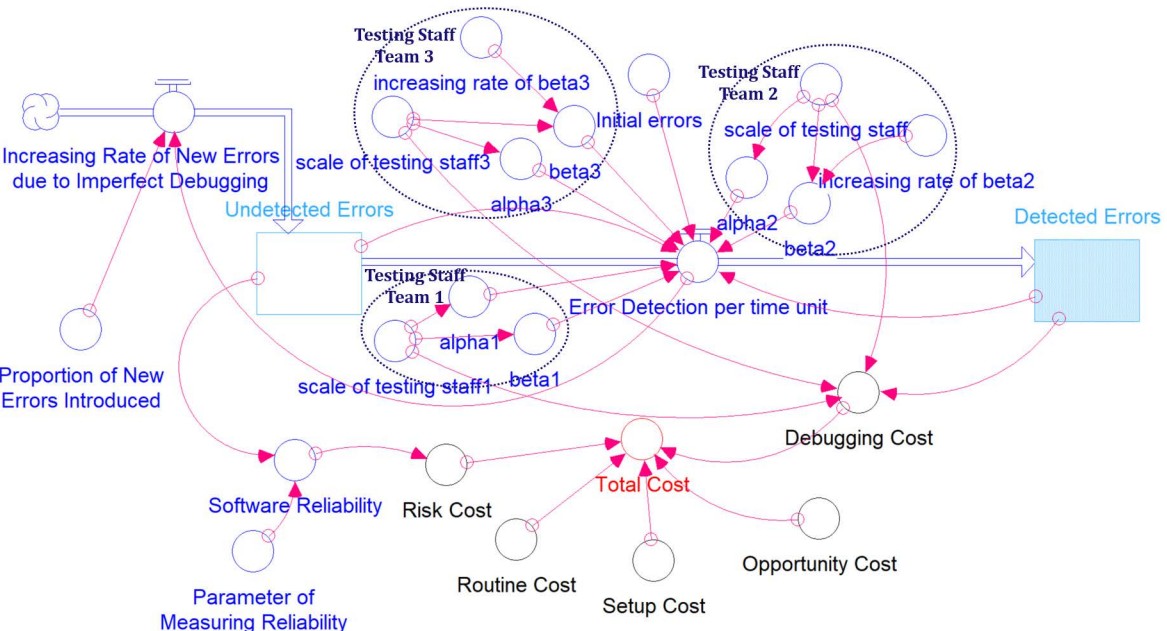

**Fig 3. Causal Loop Diagram of Basic System Dynamics Model for Software Testing Process.**

(3) Initial Errors: These represent the errors present at the beginning of the testing process. While the testing process is designed to gradually identify and reduce the number of potential hidden errors, it is important to note that the error count may not always follow a strictly decreasing trend. This is primarily due to the phenomenon of imperfect debugging, where the process of fixing existing errors can inadvertently introduce new defects or fail to fully resolve the original issues. As a result, the overall error count may experience temporary increases during the testing phase, highlighting the complex and dynamic nature of software debugging. Therefore, the initial errors can be given as a function of testing time as $A(t)$, and it will be affected by the factor "Proportion of New Errors Introduced" ($\psi$) and the flow "Increasing Rate of New Errors" ($\sum_p \psi \lambda_p(t)$). Since the flow "Increasing Rate of New Errors" may fluctuate in practice, we introduce the randomness mechanism to the flow ($\psi \sim N(\mu_\psi, \sigma_\psi)$).

(4) Increasing Rate of New Errors ($\rho$) due to Imperfect Debugging: This flow in system dynamics indicates that as the testing progresses, there is a possibility of new errors being introduced due to imperfect debugging efforts.

(5) Proportion of New Errors Introduced: Connected to the imperfect debugging process node, this represents the proportion of new errors that emerge during the debugging process.

(6) Undetected Errors: It is a stock in the system dynamics model. It means that the number of undiscovered errors decreases as the number of detected errors increases.

(7) Total Cost: This is influenced by "Setup Cost", "Routine Cost", "Debugging Cost", "Risk Cost", and "Opportunity Cost". Each of these costs is likely a factor of the resources and time invested in the testing and debugging process. The five distinct cost components, which are detailed as follows:

 (i) Setup Cost ($SC_K$): This refers to the initial investment required to organize the testing process for alternative $k$, which includes expenses for planning, acquiring equipment, and other necessary preparatory tasks.

 (ii) Routine Cost ($GC_K$): This cost accounts for ongoing operational expenses during the testing phase for alternative $k$, such as utilities, office space rental, and insurance premiums.

(iii) Debugging Cost ($DC_p$): This cost refers to the expenses associated with identifying and fixing software errors for testing staff team $p$ during the testing phase. The total debugging cost can be measured by $\sum_p DC_p M_p(t)$.

(iv) Risk Cost ($RC$): This cost quantifies the potential financial impact associated with errors that might arise after the software is released or deployed, which could result in operational disruptions or damage to the company's reputation. The cost of risk in this analysis is directly proportional to the number of errors that remain undiscovered. It is quantified here as $RC\,(1 - R(t))$.

(v) Opportunity Cost: Delaying the release of software can result in both measurable and immeasurable losses. Opportunity cost, therefore, relates to the economic impact of delays in launching the software. It is quantified here as $OC(T) = \omega_0(\omega_1 + T)^{\omega_2}$, with $\omega_0$ being a scaling factor, $\omega_1$ the initial amount, and $\omega_2$ representing how quickly the loss escalates as time progresses. The model assumes a power-law relationship for opportunity cost, which can be customized to reflect the individual perspectives or circumstances of decision-makers. This provides a versatile approach to evaluating opportunity costs in different scenarios or according to specific strategic factors.

(8) Parameter of Measuring Reliability: It denotes the symbol $\theta$, and this is likely a metric used to measure software reliability.

(9) Software Reliability: This outcome is influenced by the Undetected Errors ($A(t) - \sum_p M_p(t)$), representing the overall quality and dependability of the software after testing. It is quantified here as $R(t) = e^{-(A(t) - \sum_p M_p(t))/\theta}$. Besides, in practice, most software developers adhere to a standard of minimal software reliability ($R_m$) to ensure the quality of the software ($R(t) \geq R_m$).

Arrows represent the direction of influence between different factors indicating a positive influence. The loops, denoted by circular arrows, represent feedback processes. For example, the loop connecting Detected Errors, Undetected Errors, and Error Detection per Time Unit suggests a reinforcing loop where an increase in detected errors could lead to improved error detection efficiency, thus resulting in a higher number of detected errors in a positive feedback loop. Overall, the diagram illustrates the complex interactions among different factors in the process of software testing and debugging, the resources used, and the ultimate objective of improving software reliability.

Moreover, if the system dynamics model for software testing is large-scale, or if the manager needs to evaluate multiple distinct models for decision-making, cloud-based or distributed computing systems will be necessary to address these challenges. In scenarios involving large-scale simulations of the system dynamics, the use of cloud-based testing tools and distributed computing frameworks is essential for efficiently managing extensive datasets and enabling real-time analytics. Cloud platforms such as AWS, Azure, and Google Cloud offer scalable, on-demand computational resources that can adapt to the increasing complexity of simulations. Distributed computing technologies, such as Apache Spark and Hadoop, facilitate parallel processing across multiple nodes, significantly reducing computation time and enhancing overall efficiency. By adopting these technologies, the simulation model can not only manage larger datasets but also deliver timely and actionable results, making it a robust solution for software testing.

## 4. Application and numerical analysis

Suppose that a software development company is working on an Enterprise Resource Planning Information System (ERPIS) project. After nine months of dedicated software development, the team is now ready to enter the crucial final phase of testing and debugging. The client has set a high standard for the software's reliability, expecting it to operate with at least 90% reliability during its initial phase. This translates to a maximum of a 10% chance of encountering software errors during any given hour of operation. An analysis of the company's testing history reveals a significant impact of different testing team compositions on the efficiency of the debugging process. In this scenario, the development have the three different testing teams, and each testing team has different testing efficiencies due to their individual work

experience and ability. The three testing teams, designated as Teams 1, 2, and 3, represent varying levels of workforce intensity: low, medium, and high, respectively. Team 3, while exhibiting greater efficiency in testing compared to Teams 1 and 2, also incurs higher costs in debugging activities. Moreover, the learning factors of Teams 2 and 3 would increase with testing time to accelerate the debugging process. Here, it is assumed that the learning factors of Teams 2 and 3 increase linearly, and they can be reasonably estimated. Unfortunately, the software testing project manager cannot recruit all the members of the three teams because the software development company has another software testing project that needs to be undertaken. Therefore, the project manager for software testing can only recruit partial members from each of the three teams. Based on this situation, the project manager devises four alternatives based on the limitations of available human resources. It should be noted that the staff's debugging work is not perfect, and as a result, new errors are always introduced in software system when they attempt to correct software bugs. Furthermore, the randomness mechanism needs to incorporate the new software errors introduced in the dynamic environment. Additionally, other associated expenses also affect the overall testing cost. In this case, the setup costs and routine costs of the four alternatives are not significantly different from each other. In order to ensure software quality, the project manager set the minimum requirement for software reliability under restricted timeline for software release. Although extending software testing can effectively improve software reliability, delaying the software release may result in missed opportunities and sales loss. Suppose that the missed opportunities and sales loss can be reasonably estimated as a function. With this information, the project manager can evaluate and balance the advantages and drawbacks associated with software reliability and the costs of missed opportunities. Figure 4 and Table 4 illustrate detailed information for the four testing alternatives.

Implementing the suggested system dynamics model on the four software testing alternatives, the simulation outcomes are shown in Figures 5 and 6, and detailed in Table 5. These figures demonstrate a distinct convex correlation between the expected testing costs and the testing duration for all four alternatives. As illustrated in Figure 5, each alternative meets the minimum software reliability threshold of 90% if the testing period exceeds 5.25 months. A deeper comparative analysis indicates that alternative A4 emerges as the most economically efficient option for the organization. This is due to its lowest projected testing costs compared to the other choices. Consequently, it is prudent for the project manager to choose alternative A4, which aims to release the software after 5.6 months. At this point, the software is expected to achieve a reliability rate of 96.05%, with an estimated testing cost of $133,490. Additionally, Figure 6 reveals a noteworthy observation: the reliability trajectories of alternatives A1 and A4 are nearly identical. The progression pattern of software reliability for A1 closely mirrors that of A4, yet the costs associated with A1 are about 13–15% higher than those of A4. This finding further emphasizes the cost-effectiveness of alternative A4 in achieving comparable levels of software reliability at a reduced financial investment. Hence, this analysis offers valuable insights for decision-making in selecting the most suitable testing strategy, efficiently balancing cost and reliability. From a software quality standpoint, alternative A3 is superior to the other alternatives, even though it is not the most expensive. Should the project manager prioritize software reliability as the key factor in upholding the company's reputation, they may choose alternative A3 over A4. This choice could stem from the potential of alternative A3 to deliver superior performance, resulting in a more reliable and dependable software product. While alternative A4 may present cost advantages, alternative A3 might offer enhanced reliability measures that align more closely with the company's strategic emphasis on quality and customer trust. By choosing alternative A3, the manager demonstrates a commitment to delivering a product that not only meets, but potentially exceeds, industry standards for reliability. This decision could be crucial in strengthening the company's market position and enhancing its reputation for providing high-quality and reliable software solutions. It also reflects a strategic decision to invest in long-term brand credibility, which could potentially result in increased customer loyalty and a stronger competitive advantage. Thus, the choice of alternative A3, despite any additional costs or extended development time, could be seen as an investment in the company's reputation for excellence and reliability in software development.

In their comprehensive analysis, the project manager included additional vital factors by conducting a sensitivity analysis to explore their effects on the overall cost and the decision-making process regarding the timing of the software's

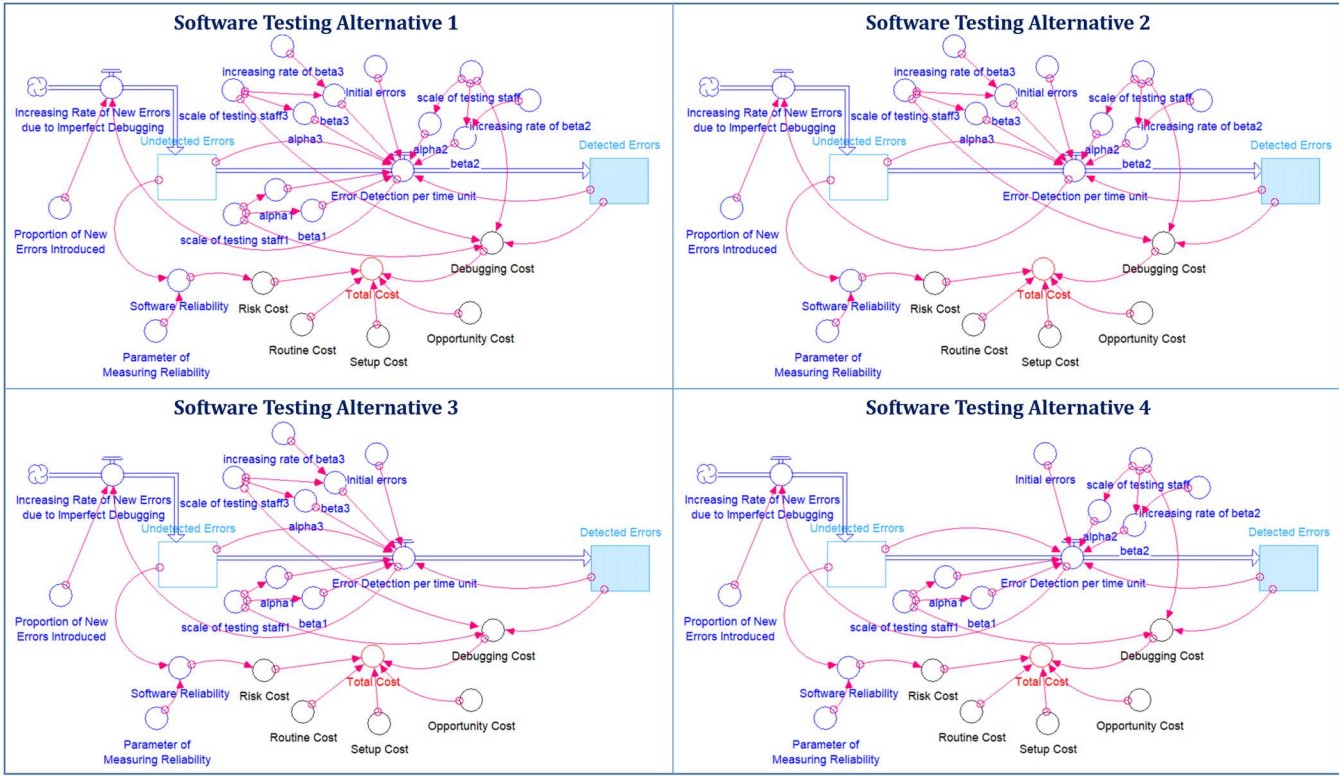

**Fig 4. Causal Loop Diagrams for Software Testing Alternatives 1–4.**

**Table 4. Related settings of system dynamics models for software testing alternatives 1–4.**

| Testing Staff Team 1 (low workforce intensity) | Testing Staff Team 2 (medium workforce intensity) | | Testing Staff Team 3 (high workforce intensity) |
|---|---|---|---|
| Parameters' value of $M_1(t)$: $\hat{\alpha}_1$=0.60, $\hat{\beta}_{10}$=1.10, $\hat{\beta}_{11}$=0 | Parameters' value of $M_2(t)$: $\hat{\alpha}_2$=0.80, $\hat{\beta}_{20}$=0.50, $\hat{\beta}_{21}$=0.025 | | Parameters' value of $M_3(t)$: $\hat{\alpha}_3$=0.90, $\hat{\beta}_{30}$=0.70, $\hat{\beta}_{31}$=0.03 |
| Debugging Cost: $DC_1$ = \$20 per error | Debugging Cost: $DC_2$ = \$25 per error | | Debugging Cost: $DC_3$ = \$35 per error |
| **Alternative 1** | **Alternative 2** | **Alternative 3** | **Alternative 4** |
| Scales of Team 1,2,3: $\xi_1$=0.4,$\xi_2$=0.3,$\xi_3$=0.3 | Scales of Team 2,3: $\xi_2$=0.5,$\xi_3$=0.5 | Scales of Team 1,3: $\xi_1$=0.3,$\xi_3$=0.7 | Scales of Team 1,2: $\xi_1$=0.7,$\xi_2$=0.3 |
| Setup Cost & Routine Cost | | | |
| $SC_1$ = \$20,500, $GC_1$ = \$4,500 | $SC_2$ = \$20,000, $GC_2$ = \$4,450 | $SC_3$ = \$20,000, $GC_3$ = \$4,450 | $SC_4$ = \$20,500, $GC_4$ = \$4,500 |
| The Parameters for Opportunity Cost: $\omega_0$ = 300, $\omega_1$ = 2.0, and $\omega_2$ = 1.9 <br> The Parameters for Risk Cost: $\theta$=50, $RC$=\$90,000 <br> All potential errors and new error introduced: <br> $A(t) = 3250 + 0.2\left(\sum_p \lambda_p(t)Rand(N(\mu_{\lambda_p(t)}, \sigma_{\lambda_p(t)}))\right)$ <br> The restricted timeline for software release: $T_U$ = 10 months <br> The minimal requirement of reliability: $R_{mr}$ = 90% | | | |

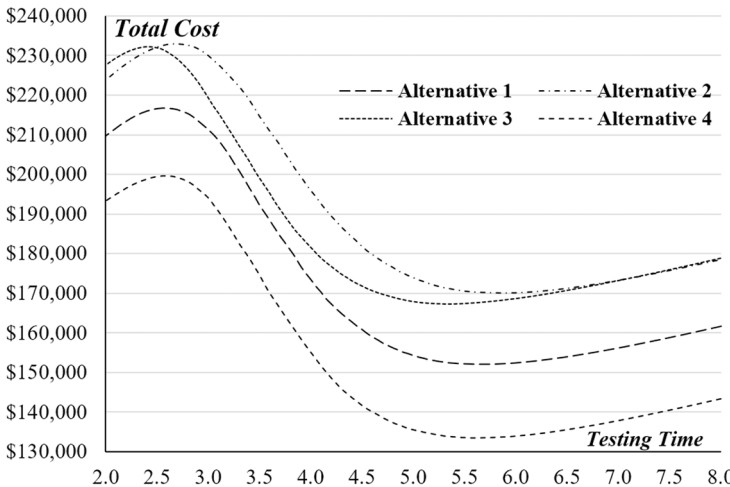

**Fig 5. Total Expected Costs versus Testing Time for the Four Alternatives.**

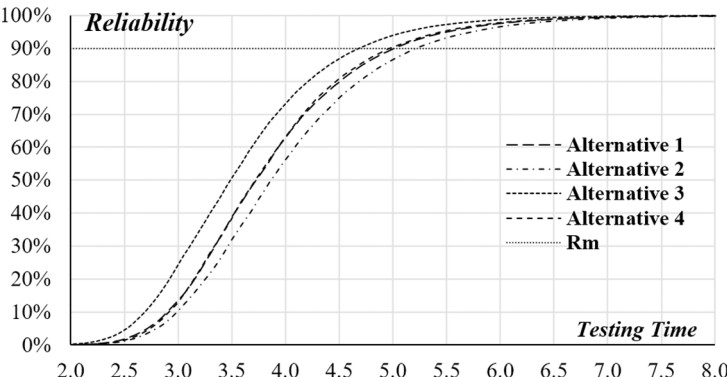

**Fig 6. Expected Reliability versus Testing Time for the Four Alternatives.**

release. Figures 7 and Table 6 detail the influences of specific parameters such as $\alpha_p$, $\beta_{p0}$, $\beta_{p1}$, $SC_K$, $GC_K$, $DC_K$, $RC$, and $OC$ on the total cost. It is evident from Figure 7 that, in terms of the model parameters, the cost shows a higher sensitivity to $\alpha_p$ and $\beta_{p0}$. This implies that inaccurate estimations of α_p and $\beta_{p0}$ can significantly disrupt the budget planning for the selected testing alternative. If the manager underestimates these parameters, there could be an overestimation of the testing costs. Additionally, incorrect predictions of $\alpha_p$ and $\beta_{p0}$ might also affect the decision regarding the software's release timing. Overestimation of these values could lead to a shortened testing phase and a rushed release, potentially resulting in customer dissatisfaction and damage to the company's reputation due to the release of unreliable software.

From another perspective, improving testing efficiency may necessitate the manager's investment in advanced training for the testing staff, consequently increasing the cost of staff education. While this investment can improve the values of $\alpha_p$ and $\beta_{p0}$, the manager must weigh the benefits of this investment against the costs associated with enhancing staff skills and the resulting decrease in expenses for reliability improvement.

Furthermore, the time-dependent routine cost ($GC_K$) and the debugging cost ($DC_K$) also play significant roles in determining the testing cost. For instance, reducing debugging costs by 10% could result in a decrease in total testing costs by approximately 6.5%. In contrast, the impact of routine costs is less significant. For example, a 10% reduction in

administrative costs may only lead to a 1% reduction in total testing costs. This finding suggests that the manager should focus on optimizing expenditures, particularly in debugging activities, to improve cost efficiency.

The study also examines the impact of random variations in error identification. Figure 8 showcases a simulation of the system dynamics, illustrating its inherent randomness. However, this randomness appears to have a limited impact on the effectiveness of testing in this particular scenario. This aspect of the analysis highlights the significance of taking into account variability and uncertainty in project management, especially in software testing, to guarantee sound decision-making and efficient resource allocation.

**Table 5. Comparative analysis of four software testing alternatives at various testing time.**

| Months | Total Cost | | | | Software Reliability | | | |
|---|---|---|---|---|---|---|---|---|
| T | A1 | A2 | A3 | A4 | A1 | A2 | A3 | A4 |
| 5.20 | $153,027 | $172,076 | $167,443 | $134,448 | 92.36% | 89.80% | 95.63% | 92.73% |
| 5.25 | $152,791 | $171,721 | $167,390 | $134,235 | 92.91% | 90.48% | 95.96% | 93.25% |
| 5.30 | $152,627 | $171,419 | $167,336 | $133,993 | 93.37% | 91.10% | 96.30% | 93.80% |
| 5.35 | $152,437 | $171,183 | **$167,319** | $133,877 | 93.87% | 91.65% | **96.60%** | 94.21% |
| 5.40 | $152,346 | $170,959 | $167,343 | $133,757 | 94.25% | 92.19% | 96.86% | 94.62% |
| 5.45 | $152,230 | $170,728 | $167,387 | $133,631 | 94.66% | 92.73% | 97.09% | 95.05% |
| 5.50 | $152,182 | $170,539 | $167,457 | $133,584 | 95.00% | 93.23% | 97.29% | 95.38% |
| 5.55 | $152,147 | $170,413 | $167,537 | $133,517 | 95.32% | 93.65% | 97.49% | 95.74% |
| 5.60 | $152,102 | $170,314 | $167,620 | **$133,490** | 95.66% | 94.05% | 97.68% | **96.05%** |
| 5.65 | **$152,096** | $170,244 | $167,723 | $133,520 | **95.95%** | 94.42% | 97.85% | 96.30% |
| 5.70 | $152,096 | $170,204 | $167,848 | $133,560 | 96.24% | 94.75% | 97.99% | 96.54% |
| 5.75 | $152,112 | $170,144 | $167,966 | $133,575 | 96.51% | 95.11% | 98.15% | 96.81% |
| 5.80 | $152,124 | $170,110 | $168,095 | $133,639 | 96.78% | 95.44% | 98.29% | 97.03% |
| 5.85 | $152,190 | **$170,079** | $168,236 | $133,713 | 97.00% | **95.77%** | 98.42% | 97.24% |
| 5.90 | $152,246 | $170,084 | $168,395 | $133,765 | 97.23% | 96.05% | 98.53% | 97.47% |
| 5.95 | $152,333 | $170,095 | $168,546 | $133,849 | 97.42% | 96.33% | 98.66% | 97.66% |
| 6.00 | $152,425 | $170,119 | $168,714 | $133,967 | 97.61% | 96.60% | 98.76% | 97.83% |

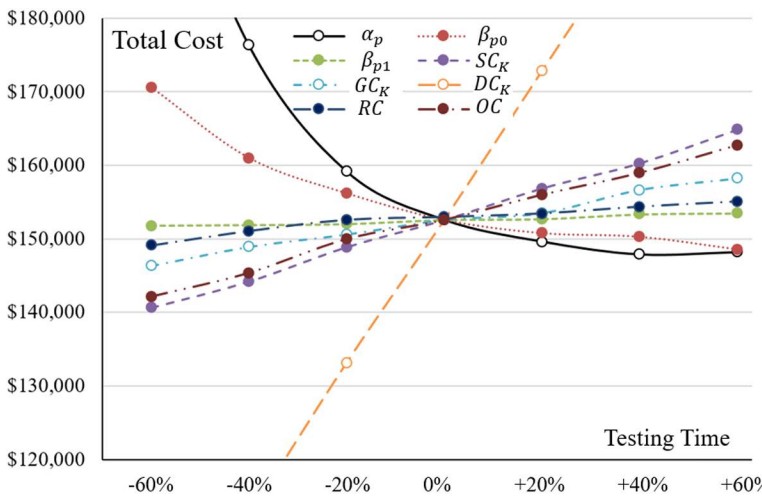

**Fig 7. The Impacts of Related Parameters on the Cost.**

## 5. Conclusions and future directions

The rapid growth of the software industry highlights the importance of a high-quality software system in enhancing a company's competitive advantage. In this context, software reliability and stability are crucial factors in the development process. Traditional software reliability growth models, although comprehensive, often involve complex mathematical formulations. Particularly in complex testing environments, these models require frequent recalibration to maintain accurate predictions, which limits their practicality in real-world applications. To address these challenges, this paper proposes the use of system dynamics to develop a mathematical model that is suitable for complex software testing scenarios. This innovative approach utilizes a system dynamics methodology to develop a software reliability growth model that effectively addresses the complexities of intricate testing environments. It involves creating a cause-and-feedback diagram and utilizing system simulation techniques to analyze the impact of testing and debugging on the growth of software reliability. This study accomplishes several objectives, as outlined below:

**Table 6. Sensitive analysis of related parameters on the cost.**

| Variation | $\alpha_p$ | $\beta_{p0}$ | $\beta_{p1}$ | $SC_K$ | $GC_K$ | $DC_K$ | $RC$ | $OC$ |
|---|---|---|---|---|---|---|---|---|
| -60% | $209,299 | $170,573 | $151,767 | $140,641 | $146,314 | $91,764 | $149,088 | $142,110 |
| -40% | $176,260 | $160,985 | $151,848 | $144,202 | $148,900 | $112,327 | $151,047 | $145,340 |
| -20% | $159,143 | $156,183 | $151,989 | $148,872 | $150,580 | $133,144 | $152,559 | $149,939 |
| 0% | $152,559 | $152,559 | $152,559 | $152,559 | $152,559 | $152,559 | $152,955 | $152,559 |
| +20% | $149,605 | $150,758 | $152,676 | $156,846 | $153,478 | $172,791 | $153,429 | $155,978 |
| +40% | $148,166 | $150,267 | $153,346 | $160,248 | $156,632 | $192,914 | $154,344 | $158,951 |
| +60% | $147,855 | $148,556 | $153,483 | $164,819 | $158,194 | $213,936 | $155,048 | $162,744 |

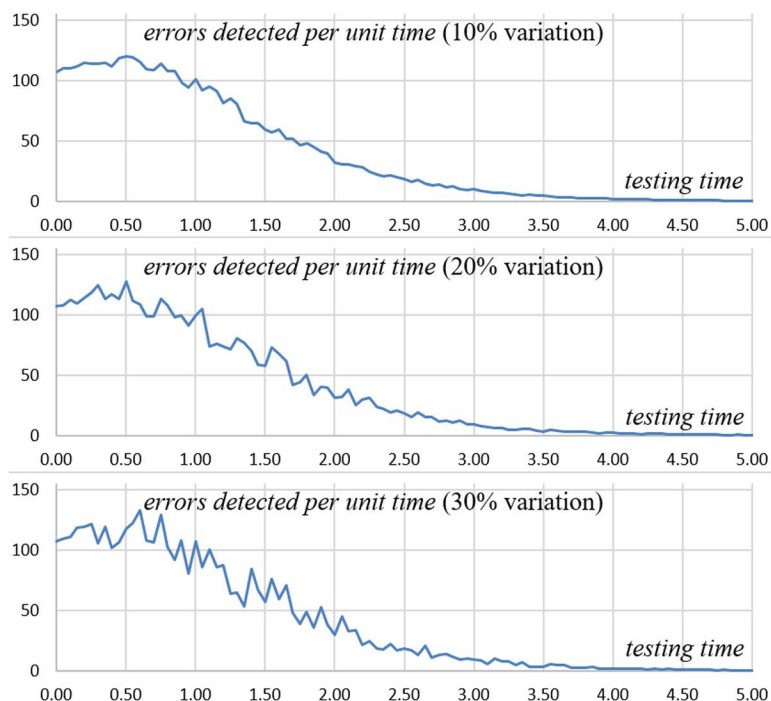

**Fig 8. The Impact of Different Random Variations on Detected Errors.**

(1) Implementation of a system dynamics approach for modeling the growth of software reliability, integrating autonomous and experiential learning factors that reflect real-world software testing dynamics. These factors, which were often overlooked in previous research, have been included in the system dynamic model to better reflect the changes in software reliability during the testing phase.

(2) Development of a system dynamics prediction model that encompasses a wide range of parameters associated with software testing and debugging costs, resulting in a comprehensive cost estimation model. This model takes into account various cost factors, enhancing the decision of budget planning and resource allocation in software development projects.

(3) An examination was conducted to assess the efficiency and cost-effectiveness of various software testing solutions. This process led to the development of individual system dynamic models for each solution. These models enable a comprehensive evaluation of the financial and reliability aspects of each testing solution, providing software developers with valuable insights to determine the most appropriate approach for their specific requirements.

Furthermore, the findings of this study provide actionable insights for both short-term and long-term project management strategies.

(1) Short-term strategies: Project managers can utilize the model to optimize the allocation of testing resources, prioritize high-impact debugging tasks, and adjust timelines based on real-time reliability predictions. For example, the cost estimation framework (Objective 2) facilitates rapid "what-if" analyses for budget adjustments during sprint cycles.

(2) Long-term strategies: Organizations can adopt the system dynamics approach to institutionalize data-driven decision-making. This may include aligning team training programs with the learning curves identified in the model (Objective 1) and establishing iterative feedback loops between testing outcomes and process improvements.

Besides, the system dynamics model introduced in this paper serves as a versatile tool for conducting sensitivity analyses on various parameters, thereby aiding in diverse decision-making processes. However, a common challenge faced in software testing is the lack of adequate historical data. This lack of data hampers the ability to accurately extract the necessary parameter values for the dynamic model, which is crucial for predicting software reliability and the associated costs. Consequently, this limitation restricts the possibility of simulating a variety of software testing scenarios, which hinders the ability of developers and decision-makers to evaluate and assess different software testing scenarios. To address these challenges, the study proposes two potential future directions.

(1) Small-sample statistics: This approach involves using a limited set of software test data to construct a smaller, yet informative sample. By doing so, it addresses the issue of insufficient historical data, enabling more accurate predictions even with limited information. This method involves utilizing advanced statistical techniques to extrapolate meaningful insights from smaller datasets, thereby enhancing the adaptability of the model in situations where extensive data is not available.

(2) Bayesian statistics and Monte Carlo simulations: This method can integrate the expertise of domain specialists with advanced computational techniques to estimate relevant parameters. Domain experts provide initial estimates, which are then refined through Bayesian statistical methods. These estimates are further enhanced using Monte Carlo simulations, which generate random samples from the parameter distributions to explore their uncertainty and variability. By iteratively sampling from these distributions, Monte Carlo methods allow for a comprehensive exploration of the parameter space, even in complex or high-dimensional scenarios. This combination of expert knowledge, Bayesian updating, and Monte Carlo sampling can be seamlessly integrated into the system dynamics model. The result is a more robust and adaptive estimation process that incorporates both subjective expertise and objective computational analysis. As more data becomes available, Bayesian statistics can further refine the parameter estimates, while Monte Carlo simulations ensure that uncertainty is explicitly accounted for. This makes the approach particularly well-suited for dynamic and uncertain environments, where flexibility and precision are critical.

In addition to these two approaches, the study can also explore other possibilities.

(1) Integration of Machine Learning: Integrating machine learning algorithms into the system dynamics model could significantly enhance its predictive accuracy and adaptability. Machine learning models, trained on both historical and ongoing software testing data, can continuously learn and adjust their predictions, thereby providing more accurate and up-to-date insights.

(2) Real-time Data Analysis: Integrating real-time data analysis into the system dynamics model can enable more agile and responsive decision-making. This approach involves continuously feeding real-time testing data into the model, enabling it to adjust its predictions based on the most recent information. As a result, it provides a more accurate and up-to-date assessment of software reliability and testing costs.

By exploring these directions, the study aims to refine and expand the applicability of the system dynamics model, making it a more effective tool for predicting software reliability and costs in diverse and data-constrained environments.

## Supporting information

**S1 Data. Failure data of Misra system 1.**
(CSV)

**S2 Data. Failure data of Misra system 2.**
(CSV)

**S3 Data. Failure data of Tandem system.**
(CSV)

**S4 Data. Failure data of Telecommunication system.**
(CSV)

## Acknowledgments

This work was sponsored by the Guangdong Basic and Applied Basic Research Foundation and the Guangdong Soft Science Foundation, China [grant number 2024A0505050040].

## Author contributions

**Conceptualization:** Wang Li, Chih-Chiang Fang.

**Funding acquisition:** Chih-Chiang Fang.

**Methodology:** Wang Li, Chih-Chiang Fang.

**Project administration:** Chih-Chiang Fang.

**Software:** Wang Li.

**Supervision:** Chih-Chiang Fang.

**Writing – original draft:** Wang Li, Chih-Chiang Fang.

**Writing – review & editing:** Wang Li, Chih-Chiang Fang.

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
