## [Decision Letter · Decision Letter 0]

4 Feb 2025

PONE-D-24-47186Applying a System Dynamics Approach for Decision-Making in Software Testing ProjectsPLOS ONE

Dear Dr. Fang,

Thank you for submitting your manuscript to PLOS ONE. After careful consideration, we feel that it has merit but does not fully meet PLOS ONE’s publication criteria as it currently stands. Therefore, we invite you to submit a revised version of the manuscript that addresses the points raised during the review process. Please note that the reviewer suggested some references to be included. Please cite the references only when you find them relevant with the topic of the manuscript.

We look forward to receiving your revised manuscript.

Kind regards,

Iftikhar Ahmed Khan

Academic Editor

PLOS ONE

“This work was sponsored by the Guangdong Basic and Applied Research Foundation, China [grant number 2024A0505050040]”

Reviewers' comments:

Reviewer's Responses to Questions

**Comments to the Author**

1. Is the manuscript technically sound, and do the data support the conclusions?

Reviewer #1: Yes

Reviewer #2: Yes

2. Has the statistical analysis been performed appropriately and rigorously? 

Reviewer #1: Yes

Reviewer #2: Yes

3. Have the authors made all data underlying the findings in their manuscript fully available?

Reviewer #1: No

Reviewer #2: No

4. Is the manuscript presented in an intelligible fashion and written in standard English?

Reviewer #1: Yes

Reviewer #2: Yes

5. Review Comments to the Author

Reviewer #1: 1- The introduction establishes the importance of software quality, but it needs to highlight the contribution of your study.

2- Consider breaking up longer sentences for improved readability. For instance, the sentence starting with "However, the challenging balance..." could be split for clarity.

3- Figure 1 is the good addition.

4- You mention that traditional models are complex and impractical. Expanding on how your model specifically addresses these shortcomings would strengthen your argument and provide a clearer rationale for your approach.

5- Paper working on Decision-Making in Software Testing Projects, so some similar work may be considered in the introduction section and also in total expected cost analysis part.

(i) Pradhan, V., Dhar, J., & Kumar, A. (2022). Software reliability models and multi-attribute utility function based

strategic decision for release time optimization. In Predictive Analytics in System Reliability (pp. 175-190).

Cham: Springer International Publishing.

(ii) Pradhan, V., Kumar, A., & Dhar, J. (2023). Emerging trends and future directions in software reliability growth

modeling. Engineering reliability and risk assessment, 131-144.

6- In Subsection 2.3 Model Verification and Comparison, more comparison methods are required like predictive power (PP) and predictive risk ratio (PRR). See the comparison of R-square of proposed model and Pham and Zhang [5] are same for all the datasets.

7- Your conclusion touches on the impact of your research on decision-making. Strengthen this section by discussing potential implications for project management and strategies in both the short and long term.

Reviewer #2: After a detailed analysis of the uploaded document, here are six critical recommendations to address the identified shortcomings, along with relevant research papers for further exploration.

The study presents theoretical models for software reliability but lacks practical deployment scenarios. Include real-world case studies demonstrating the model's application in diverse industries, such as financial systems or healthcare software.

While the paper explores mathematical models, it assumes "perfect debugging," which is often unrealistic. Discuss how factors like imperfect debugging and evolving software environments impact the model's effectiveness. Incorporate contemporary approaches such as dynamic fault modeling or machine learning-driven predictions.

Most modern software development adopts Agile methodologies, but the study does not address iterative testing scenarios. Suggest adaptations of the proposed models for continuous integration and deployment pipelines.

The manuscript evaluates the proposed model but does not adequately benchmark it against state-of-the-art reliability growth models (SRGMs). Include comparative results with models like NHPP and FeSAD frameworks to validate the model’s superiority.

The reliance on maximum likelihood estimation (MLE) and least squares estimation (LSE) is limiting. Introduce additional statistical techniques like Bayesian inference or Monte Carlo simulations to improve parameter estimation accuracy under uncertain data conditions.

The proposed approach should explicitly discuss scalability for large-scale distributed systems. Suggest integrating cloud-based testing tools or distributed computing to handle extensive datasets and real-time analytics.

Update your literature and study the following papers:

1- https://scholar.google.com/citations?view_op=view_citation&hl=en&user=Ztcq8g8AAAAJ&citation_for_view=Ztcq8g8AAAAJ:hqOjcs7Dif8C

2- https://scholar.google.com/citations?view_op=view_citation&hl=en&user=Ztcq8g8AAAAJ&citation_for_view=Ztcq8g8AAAAJ:IWHjjKOFINEC

3- https://scholar.google.com/citations?view_op=view_citation&hl=en&user=Ztcq8g8AAAAJ&citation_for_view=Ztcq8g8AAAAJ:W7OEmFMy1HYC

6. PLOS authors have the option to publish the peer review history of their article (what does this mean? ). If published, this will include your full peer review and any attached files.

**Do you want your identity to be public for this peer review?** For information about this choice, including consent withdrawal, please see our Privacy Policy .

Reviewer #1: **Yes: ** Vishal Pradhan

Reviewer #2: **Yes: ** Dr.Islam Zada

---

## [Author Response · Author response to Decision Letter 1]

17 Mar 2025

The Detailed Response to Reviewers (and Editors)

Manuscript ID: PONE-D-24-47186

Title: Applying a System Dynamics Approach for Decision-Making in Software Testing Projects

Dear Editor-in-Chief,

Thanks for the insightful and constructive comments made by the reviewers. The manuscript has been substantially revised according to the reviewers’ comments, and the following pages summarize our explanations and the revision works. Note that all the revisions are marked with red texts in the manuscript.

Below is a summary of the key points raised by the reviewers and our responses:

Reviewer #1:

1. Highlighting Contributions: We have revised the last two paragraphs of the introduction to explicitly outline the three key contributions of our study: (1) integrating dynamic interactions among testing, debugging, and resource allocation; (2) enhancing practical relevance by incorporating real-world factors; and (3) accounting for the learning curve of personnel.

2. Improving Readability: We have divided longer sentences throughout the manuscript, especially in the abstract and key sections, to improve clarity and flow.

3. Model Advantages: We have expanded the discussion on how our system dynamics model addresses the limitations of traditional models, especially in scenarios involving multiple testing teams and imperfect debugging.

4. Model Verification: We explained why metrics like predictive power (PP) and predictive risk ratio (PRR) are not suitable for our study due to the characteristics of time-series software testing data. Additionally, we corrected editorial errors in Table 3.

5. Strengthening Conclusions: We enhanced the conclusion by discussing both the short-term and long-term implications of our research for project management, particularly in relation to resource allocation and data-driven decision-making.

Reviewer #2:

1. Practical Deployment Scenarios: Although our primary focus is on methodological development, we have included a detailed hypothetical example in Section 4 to demonstrate the practical application of our model in a real-world project setting.

2. Imperfect Debugging: We modified the model to accommodate imperfect debugging by integrating stochastic elements into the time-dependent function A(t), thereby enhancing the simulation's alignment with real-world practices.

3. Agile Methodologies: We recognize the significance of Agile practices; however, we clarified that incorporating them into our existing framework necessitates additional research. We intend to investigate this further in future studies.

4. Benchmarking: We clarified our selection of NHPP-based models for benchmarking and explained why FeSAD frameworks are not directly applicable to our study.

5. Statistical Techniques: We have included a discussion on the potential application of Bayesian inference and Monte Carlo simulations in future research to enhance parameter estimation under conditions of uncertain data.

6. Scalability: We have included a discussion on how our model can be adapted for large-scale distributed systems, including the potential integration of cloud-based testing tools and distributed computing.

Reviewer #1:

Questions:

1. The introduction establishes the importance of software quality, but it needs to highlight the contribution of your study.

REPLY:

Thank you for your suggestion. We have rewritten the last two paragraphs of the introduction to highlight the contributions of our study. The content of the revised paragraphs is as follows:

The study aims to integrate real-world scenarios into the system dynamics model, thereby providing a more realistic and practical perspective. This includes considering factors such as team capabilities, resource limitations, and market pressures, all of which can significantly impact the reliability of software and the testing processes. In summary, the study introduces an innovative framework for decision-making in software testing projects by applying a system dynamics approach, offering three key contributions to the field. First, the proposed methodology presents a comprehensive scientific management framework that integrates dynamic interactions among software testing, debugging processes, and resource allocation. Second, the model enhances practical relevance by incorporating a broader array of real-world factors, such as evolving team dynamics, imperfect debugging scenarios, and changing project constraints. Third, the study also accounts for the learning curve of testing and debugging personnel, allowing for a more accurate representation of improvements in software reliability throughout the testing process.

We hope that the revised texts can address your concerns. If there are any deficiencies, we can make further improvements.

Please see page 3 for the revision.

2. Consider breaking up longer sentences for improved readability. For instance, the sentence starting with “However, the challenging balance...” could be split for clarity.

REPLY:

Thank you for your valuable feedback and constructive suggestions. I have carefully reviewed your comment regarding the readability of longer sentences, particularly the example you provided starting with “However, the challenging balance...”. In response, I have made extensive revisions throughout the manuscript, including breaking up longer sentences into shorter, more concise ones to enhance clarity and flow. I hope these changes address your concern and improve the overall readability of the paper.

I appreciate your time and effort in helping me refine this work. Please let me know if there are any additional adjustments needed.

Please see the abstract and pages 2, 4, 7, and 18 for the revisions.

3. Figure 1 is the good addition. You mention that traditional models are complex and impractical. Expanding on how your model specifically addresses these shortcomings would strengthen your argument and provide a clearer rationale for your approach.

REPLY:

Thank you for your questions. In traditional models, all testing teams are treated as a single entity, and the parameters of the software reliability growth model are estimated based on this collective unit. However, in practice, testing teams often undergo changes and reorganization. In such cases, the model parameters previously estimated for the unified team are no longer applicable and must be re-estimated, as the values of these parameters have shifted. Furthermore, when considering multiple testing teams and the possibility of imperfect testing introducing new software defects, it becomes challenging to estimate testing efficiency using past mathematical inference methods. This is because deriving a closed-form mathematical model under these conditions is highly complex. Please refer to Figure A, which illustrates that each testing team’s testing efficiency can be described using a mathematical model. However, when considering the scenario of imperfect debugging across all teams, the combined mathematical models do not yield a closed-form expression to fully characterize the entire situation. To address this limitation, we utilize a system dynamics model, which allows for a comprehensive and dynamic representation of the entire testing process.

We have addressed these issues in the revised manuscript, and we hope that the modifications provide readers with a clearer understanding of the subject.

Figure A. Causal Loop Diagram Model with Multiple Teams for Software Testing Process

Please refer to page 10 for the revisions.

4. Paper working on Decision-Making in Software Testing Projects, so some similar work may be considered in the introduction section and also in total expected cost analysis part.

(i) Pradhan, V., Dhar, J., & Kumar, A. (2022). Software reliability models and multi-attribute utility function based strategic decision for release time optimization. In Predictive Analytics in System Reliability (pp. 175-190).

Cham: Springer International Publishing.

(ii) Pradhan, V., Kumar, A., & Dhar, J. (2023). Emerging trends and future directions in software reliability growth modeling. Engineering reliability and risk assessment, 131-144.

REPLY:

We sincerely thank the reviewer for recommending the relevant literature. These references have effectively addressed the gaps identified in our Introduction, enriching the depth and comprehensiveness of our literature review. We have incorporated the suggested references into the manuscript and updated the Reference list accordingly.

Please refer to pages 2 and 21 for the revisions.

5. In Subsection 2.3 Model Verification and Comparison, more comparison methods are required like predictive power (PP) and predictive risk ratio (PRR). See the comparison of R-square of proposed model and Pham and Zhang [5] are same for all the datasets.

REPLY:

Thank you for your questions. To evaluate metrics such as predictive power (PP) and predictive risk ratio (PRR), it is essential to divide the original data into two distinct sets: the model construction data (training data) and the testing data. This division is crucial for ensuring the validity and reliability of the predictive metrics.

However, in publicly available software testing datasets, the data consists of time-series records of testing activities. Extracting any segment of this time-series data for testing purposes would create a gap in the training data, leading to significant bias in the model parameters. Given this characteristic of software testing data, metrics such as predictive power (PP) and predictive risk ratio (PRR) are not suitable for our research problem. This is why these two metrics are rarely used to evaluate predictive performance in related studies. Instead, most research in this field relies on measures of fitting ability, such as mean squared error (MSE) and R-squared, to assess model performance. For this reason, our study adopts the evaluation metrics commonly used in the majority of related research.

Besides, thank you for bringing the issue regarding Table 3 to our attention. In the previous version, there were some editorial errors in Table 3, which have now been corrected. Additionally, the identical R-square values observed across different models are due to the precision being limited to three decimal places.

Please refer to Table 3 on page 9 for the revisions.

6. Your conclusion touches on the impact of your research on decision-making. Strengthen this section by discussing potential implications for project management and strategies in both the short and long term.

REPLY:

Thank you for your constructive feedback on strengthening the discussion of managerial implications. We have revised the Conclusions section to explicitly address short-term and long-term project management strategies, including:

(1) Short-term strategies: Project managers can utilize the model to optimize the allocation of testing resources, prioritize high-impact debugging tasks, and adjust timelines based on real-time reliability predictions. For example, the cost estimation framework (Objective 2) facilitates rapid “what-if” analyses for budget adjustments during sprint cycles.

(2) Long-term strategies: Organizations can adopt the system dynamics approach to institutionalize data-driven decision-making. This may include aligning team training programs with the learning curves identified in the model (Objective 1) and establishing iterative feedback loops between testing outcomes and process improvements.

Please refer to page 19 for the revisions.

Reviewer #2:

General Comment: After a detailed analysis of the uploaded document, here are six critical recommendations to address the identified shortcomings, along with relevant research papers for further exploration.

Questions:

1. The study presents theoretical models for software reliability but lacks practical deployment scenarios. Include real-world case studies demonstrating the model’s application in diverse industries, such as financial systems or healthcare software.

REPLY:

Thank you for your valuable feedback regarding the inclusion of practical deployment scenarios in our study. We appreciate your suggestion to incorporate real-world case studies from diverse industries, such as financial systems or healthcare software, to demonstrate the model’s applicability.

However, we would like to clarify that the primary focus of this research is to propose a methodological framework that addresses the limitations of traditional models in estimating software reliability and testing costs, particularly in scenarios involving multiple testing teams. This methodology is designed to provide software testing managers with a flexible and adaptive tool for effective resource allocation to achieve project goals.

To illustrate the practical application of our approach, we included a detailed example in Section 4 (Application and Numerical Analysis). In this section, we modeled a scenario where a software development company is tasked with developing an Enterprise Resource Planning Information System (ERPIS) project. The example demonstrates how our methodology can be used to formulate a software testing plan, optimize resource allocation, and predict reliability and costs in a real-world project setting. While this example is hypothetical, it is grounded in realistic assumptions and reflects the challenges faced by software testing managers in practice.

We acknowledge that including actual deployment data from real-world projects, such as financial or healthcare systems, could further strengthen the persuasiveness of our findings. However, as our research primarily emphasizes the development and application of the methodology, we have not sought to publicly disclose proprietary data from specific companies. We believe that the hypothetical example provided in Section 4 effectively showcases the practicality and relevance of our approach without compromising the confidentiality of real-world data.

We hope that the example in Section 4, along with the methodological contributions of our study, sufficiently demonstrates the value and applicability of our framework. We kindly request to retain the current content, as it aligns with the research’s focus on advancing a flexible and adaptive methodology for software reliability and testing cost estimation.

Thank you again for your thoughtful comments. Please let us know if further clarifications or adjustments would enhance the manuscript.

2. While the paper explores mathematical models, it assumes “perfect debugging,” which is often unrealistic. Discuss how factors like imperfect debugging and evolving software environments impact the model’s effectiveness. Incorporate contemporary approaches such as dynamic fault modeling or machine learning-driven predictions.

REPLY:

Thank you for your insightful questions. While the foundational mathematical model was initially derived under the assumption of perfect debugging, our subsequent derivations allow for the transformation of the number of initial errors (A) into a time-dependent function (A(t)). This enables the conversion of the original perfect debugging Software Reliability Growth Model (SRGM) into an imperfect debugging SRGM. We apologize for not clearly explaining this in the previous version of the manuscript. In this revised version, we have added and modified relevant discussions to ensure clarity for readers. Furthermore, in our system dynamics model, we have incorporated stochastic elements into the function A(t). Although A(t) increases over time, it follows a probability distribution, making the simulation scenarios more aligned with real-world practices.

Regarding your mention of machine learning-driven predictions, we acknowledge their effectiveness in predicting software reliability. However, since our methodology is based on system dynamics, integrating machine learning approaches into our model is not straightforward. In future research, we aim to explore ways to combine system dynamics with machine learning, which could potentially enhance the predictive accuracy of the model.

Please refer to pages 5, 6, 7, 11, and 12 for the revisions.

3. Most modern software development adopts Agile methodologies, but the study does not address iterative testing scenarios. Suggest adapt

---

## [Decision Letter · Decision Letter 1]

15 Apr 2025

Applying a System Dynamics Approach for Decision-Making in Software Testing Projects

PONE-D-24-47186R1

Dear Dr. Fang,

We’re pleased to inform you that your manuscript has been judged scientifically suitable for publication and will be formally accepted for publication once it meets all outstanding technical requirements.

Kind regards,

Iftikhar Ahmed Khan

Academic Editor

PLOS ONE

Additional Editor Comments (optional):

Reviewers' comments:

Reviewer's Responses to Questions

**Comments to the Author**

1. If the authors have adequately addressed your comments raised in a previous round of review and you feel that this manuscript is now acceptable for publication, you may indicate that here to bypass the “Comments to the Author” section, enter your conflict of interest statement in the “Confidential to Editor” section, and submit your "Accept" recommendation.

Reviewer #1: All comments have been addressed

2. Is the manuscript technically sound, and do the data support the conclusions?

Reviewer #1: Yes

3. Has the statistical analysis been performed appropriately and rigorously? 

Reviewer #1: I Don't Know

4. Have the authors made all data underlying the findings in their manuscript fully available?

Reviewer #1: Yes

5. Is the manuscript presented in an intelligible fashion and written in standard English?

Reviewer #1: Yes

6. Review Comments to the Author

Reviewer #1: Authors have incorporated all the suggestions.

7. PLOS authors have the option to publish the peer review history of their article (what does this mean? ). If published, this will include your full peer review and any attached files.

**Do you want your identity to be public for this peer review?** For information about this choice, including consent withdrawal, please see our Privacy Policy .

Reviewer #1: **Yes: ** Vishal Pradhan

---

## [Editor Report · Acceptance letter]

PONE-D-24-47186R1

PLOS ONE

Dear Dr. Fang,

I'm pleased to inform you that your manuscript has been deemed suitable for publication in PLOS ONE. Congratulations! Your manuscript is now being handed over to our production team.

Kind regards,

on behalf of

Dr. Iftikhar Ahmed Khan

Academic Editor

PLOS ONE